# Timely and Personalized Interventions and Vigilant Care in Neurodegenerative Conditions: The FIT4TeleNEURO Pragmatic Trial

**DOI:** 10.3390/healthcare13060682

**Published:** 2025-03-20

**Authors:** Francesca Baglio, Federica Rossetto, Elisa Gervasoni, Ilaria Carpinella, Giulia Smecca, Irene Aprile, Roberto De Icco, Stefania De Trane, Chiara Pavese, Christian Lunetta, Cira Fundarò, Laura Marcuccio, Giovanna Zamboni, Franco Molteni, Cristina Messa

**Affiliations:** 1IRCCS Fondazione Don Carlo Gnocchi ONLUS, 20148 Milan, Italy; fbaglio@dongnocchi.it (F.B.); egervasoni@dongnocchi.it (E.G.); icarpinella@dongnocchi.it (I.C.); cristina.messa@unimib.it (C.M.); 2IRCCS Fondazione Don Carlo Gnocchi ONLUS, 50143 Florence, Italy; iaprile@dongnocchi.it; 3Department of Brain and Behavioral Sciences, University of Pavia, 27100 Pavia, Italy; roberto.deicco@mondino.it; 4Movement Analysis Research Section, IRCCS Mondino Foundation, 27100 Pavia, Italy; 5Istituti Clinici Scientifici Maugeri IRCCS, 70124 Bari, Italy; stefania.detrane@icsmaugeri.it; 6Department of Clinical-Surgical, Diagnostic and Pediatric Sciences, University of Pavia, 27100 Pavia, Italy; chiara.pavese@unipv.it; 7Istituti Clinici Scientifici Maugeri IRCCS, Centro Studi Attività Motorie and Neurorehabilitation and Spinal Units of Pavia Institute, 27100 Pavia, Italy; 8Istituti Clinici Scientifici Maugeri IRCCS, Neurorehabilitation Unit of Milan Institute, 20138 Milan, Italy; christian.lunetta@icsmaugeri.it; 9Istituti Clinici Scientifici Maugeri IRCCS, 27040 Montescano, Italy; cira.fundaro@icsmaugeri.it; 10Istituti Clinici Scientifici Maugeri IRCCS, 82037 Telese, Italy; laura.marcuccio@icsmaugeri.it; 11Dipartimento di Scienze Biomediche, Metaboliche e Neuroscienze, Università di Modena e Reggio Emilia, 41125 Modena, Italy; giovanna.zamboni@unimore.it; 12Centro di Riabilitazione “Villa Beretta”, Ospedale Valduce, 23845 Costa Masnaga, Italy; fmolteni@valduce.it; 13School of Medicine and Surgery, University of Milano-Bicocca, 20126 Milan, Italy

**Keywords:** digital medicine, telemedicine, telerehabilitation, rehabilitation, chronic neurological diseases, Parkinson’s disease, multiple sclerosis

## Abstract

Parkinson’s disease (PD) and multiple sclerosis (MS) are two chronic neurological diseases (CNDs) that have a high demand for early and continuous rehabilitation. However, accessing professional care remains a challenge, making it a key priority to identify sustainable solutions for ensuring early rehabilitation availability. **Objective**: The FIT4TeleNEURO pragmatic trial proposes to investigate, in real-life care settings, the superiority in terms of the effectiveness of early rehabilitation intervention with harmonized, mix-model telerehabilitation (TR) protocols (TR single approach, task-oriented—TRsA; TR combined approach, task-oriented and impairment-oriented—TRcA) compared to conventional management (control treatment, CeT) in people with PD and MS. **Design, and Methods**: This multicenter, randomized, three-treatment arm pragmatic trial will involve 300 patients with CNDs (PD, N = 150; MS, N = 150). Each participant will be randomized (1:1:1) to the experimental groups (20 sessions of TRsA or TRcA according to a mix-model—3 asynchronous + 1 synchronous session/week) or the control group (20 sessions of CeT). Primary and secondary outcome measures will be obtained at the baseline (T0), post-intervention (T1, 5 weeks after baseline), and follow-up (T2, 3 months after the end of the treatment). A multidimensional evaluation (cognitive, motor, and quality of life domains) will be conducted at each time point of assessment (T0; T1; T2). The primary outcome measures will be the assessment of change (T0 vs. T1 vs. T2) in static and dynamic balance, measured using the Mini-Balance Evaluation Systems Test. Usability and acceptability assessment will be also investigated. **Expected Results**: Implementing TR protocols will enable a more targeted and efficient response to the growing demand for rehabilitation in the early stages of CNDs. Both the TRsA and TRcA approaches are expected to be more effective than CeT, with the combined approach likely providing greater benefits in secondary outcome measures. Finally, the acceptability of the asynchronous modality could open the door to scalable solutions, such as digital therapeutics.

## 1. Introduction

Chronic neurological disorders (CNDs) are the second leading cause of death worldwide [1]. Due to their progressive nature and neurodegenerative characteristics, these disorders often lead to long-term disabilities that affect physical, cognitive, behavioral, and social functioning [2]. While pharmacological interventions play a crucial role, rehabilitation strategies are essential for managing disability, particularly when implemented from the early stages of the disease.

Parkinson’s disease (PD) and multiple sclerosis (MS) are among the CND conditions with the highest demand for rehabilitation. Despite differing in etiopathogenetic mechanisms, both disorders share a progressive neurodegenerative course and common rehabilitative objectives, such as improving functional mobility and physical capacity [3,4,5,6]. These areas benefit significantly from rehabilitation, especially when treatment is initiated early and delivered intensively [7,8,9,10]. Early, intensive, and accessible interventions have been shown to enhance exercise tolerance, walking ability, and balance [11,12,13], while significantly reducing the risk of falls, a critical concern for these patients [14,15,16].

As highlighted by Cieza and colleagues [17] in their analysis of the Global Burden of Diseases, Injuries, and Risk Factors Study (GBD 2019), an estimated 2.41 billion individuals (95% uncertainty interval 2.34–2.50 billion) worldwide could benefit from rehabilitation services. This vision challenges the misconception that rehabilitation is a niche service required by only a small segment of the population. The authors further argue that rehabilitation must be integrated into primary healthcare systems to ensure equitable access, particularly for underserved communities. In the context of the Italian National Health Service, equitable access to rehabilitation remains a significant challenge. Patients in the early stages of chronic disability are particularly affected, as no standardized care pathways exist to ensure timely and continuous management of their needs. Addressing this gap by developing sustainable and effective solutions for early access to rehabilitation services for CND patients is an urgent priority.

Technological advancements in healthcare, as highlighted by Topol [18], provide an opportunity to transform the healthcare system and care delivery, with digital medicine and telemedicine playing a pivotal role in this transition. Among telemedicine applications, telerehabilitation (TR) has emerged as a promising approach for delivering integrated, continuous care for individuals with chronic neurological disabilities from the earliest stages [19,20,21]. Research has demonstrated the feasibility of TR interventions for various CNDs, including MS [22,23,24,25,26,27,28] and PD [29,30,31,32,33,34,35,36,37], as well as conditions within the Alzheimer’s disease continuum [38,39]. These studies suggest that TR can achieve clinical outcomes and medical benefits comparable to traditional rehabilitation methods. However, limitations such as heterogeneous intervention protocols, small sample sizes, and a lack of standardized approaches remain challenges.

Task-oriented training is the predominant rehabilitation approach employed in the early stages of CNDs to enhance functional mobility and physical capacity. Additionally, growing evidence suggests that integrating task-oriented strategies with impairment-oriented activities may result in superior outcomes [40,41,42]. However, there remains a lack of standardized approaches in both TR and conventional care, underscoring the urgent need to establish consistent and evidence-based protocols.

The FIT4TeleNEURO study aims to implement a harmonized, multicenter TR protocol specifically designed for the early stages of PD and MS to address gait and balance difficulties at home. The study will evaluate whether this TR protocol is acceptable and effective through a pragmatic trial design, assessing its superiority over conventional care in early rehabilitative management. Furthermore, secondary objectives include examining the specific effects of rehabilitative strategies by comparing task-oriented and combined task- and impairment-oriented approaches on motor, cognitive, and quality-of-life outcomes, as well as investigating the differential impact of these digital interventions on MS and PD cohorts.

## 2. Materials and Methods

The study protocol has been developed in accordance with the guidelines outlined in the Standard Protocol Items: Recommendations for Interventional Trials (SPIRIT) (Figure 1). The study will be conducted according to the Declaration of Helsinki 2024 [43].

### 2.1. Trial Design and Setting

This pragmatic study is designed as a single-blinded, randomized, three-treatment-arm (1:1:1) study involving chronic outpatients from neurorehabilitation units of 10 FIT4TeleNEURO Centers: (1) IRCCS Fondazione Don Carlo Gnocchi ONLUS, Milano (MI); (2) IRCCS Fondazione Don Carlo Gnocchi ONLUS, Roma (RO); (3) IRCCS Fondazione Mondino, Pavia (PV); (4) IRCCS Istituti Clinici Scientifici Maugeri, Bari (BA); (5) IRCCS Istituti Clinici Scientifici Maugeri, Milano (MI); (6) IRCCS Istituti Clinici Scientifici Maugeri, Pavia (PV); (7) IRCCS Istituti Clinici Scientifici Maugeri, Montescano (PV); (8) IRCCS Istituti Clinici Scientifici Maugeri, Telese (BN); (9) Università di Modena e Reggio Emilia, Modena (MO); and (10) Ospedale Valduce, Centro di Riabilitazione “Villa Beretta”, Como (CO).

Following enrollment and the baseline assessment, each participant will be randomly assigned into one of the following three groups, with an allocation ratio of 1:1:1: (1) telerehabilitation single approach (TRsA, experimental group); (2) telerehabilitation combined approach (TRcA, experimental group); or (3) conventional educational treatment (CeT, placebo comparator).

Randomization will be stratified based on both the clinical center and the clinical conditions (MS, PD). Primary and secondary outcome measures will be collected at baseline (T0), post-intervention (T1, 5 weeks after baseline), and at follow-up (T2, 12 weeks after treatment completion). The trial work plan is shown in Figure 2.

### 2.2. Sample Size

The sample size was determined a priori using G*Power 3 software, based on previously published preliminary data [30] related to the study’s primary outcome, the Mini-Best test. Assuming a minimal clinically important difference (MCID) of 3 points (ds group 1 = 5.75, ds_group 2 = 5.98), a statistical power of 80% (type I error rate of 0.05), and an allocation ratio of 1:1:1, 48 subjects per group are required. These groups include TR with single approach (TRsA) and the control group. A third group (TR with combined approach, TRcA) of 48 subjects will be recruited to assess whether a neuromotor rehabilitation program combining two approaches (task-oriented and impairment-oriented) is superior to the control group and differs from the TRsA group in secondary outcomes. The study will recruit 144 patients with Parkinson’s disease and 144 patients with multiple sclerosis. To account for a potential 5% drop-out rate per pathology group, a total of 300 subjects will be enrolled.

### 2.3. Study Population, Recruitment, and Randomization

Based on the sample size calculation, the trial intends to recruit 300 individuals with CND, comprising 150 diagnosed with PD and 150 with MS. Recruitment will be competitive across participating centers. Eligible patients who meet all inclusion criteria (outlined in the section below) will be randomized through a web-based allocation concealment method, using a computer algorithm created by an independent statistician. Stratified randomization will be utilized to prevent imbalances between treatment groups. Participants will be stratified according to their clinical center and condition (MS, PD) and then randomly assigned (1:1:1) to one of the experimental groups (TRsA or TRcA) or the control group (CeT) (Figure 3). The intervention will not be blinded for either clinicians or patients due to its nature. However, clinical endpoints and data collection from clinical/psychological questionnaires will be blinded for examiners/assessors. The statistician conducting the data analysis will also be blinded to group allocation.

### 2.4. Inclusion and Exclusion Criteria

Given the pragmatic nature of the study, the inclusion criteria for all participants will be:Diagnosis of probable PD according to MDS criteria [44] in staging between 1 and 3 on the Hoehn and Yahr scale [45] or diagnosis of MS according to the 2017 revised criteria of MC Donald [46] with disability level at the Expanded Disability Status Scale EDSS [47] ≤ 4.5.Age between 25 and 85 years.The preserved cognitive level at the Montreal Cognitive Assessment test (MoCA test > 15.5) [48].No rehabilitation program being implemented at the time of enrollment.Stable drug treatment (last three months) with L-Dopa or dopamine agonists (PD group) or disease modifying therapies (DMTs) (MS group).

The exclusion criteria for all participants will be:The presence of comorbidities that could hinder patients from safely participating in a home program or indicate clinical instability (e.g., severe orthopedic issues or significant cognitive impairments).Unsuitable environmental factors, such as inadequate space for rehabilitation activities or the absence of a stable internet connection.The presence of major psychiatric complications or personality disorders, assessed through a clinical interview.The presence of severe impairments in visual and/or auditory perception.Falls resulting in injuries or more than two falls in the six months prior to recruitment (for both PD and MS groups).Relapse ongoing/less than 3 months since the last relapse (MS group).The presence of “frequent” freezing as recorded at the administration of Section II (daily life activity) of the UPDRS (score ≥ 3) (PD group).EDSS-FS (cerebellar function) ≥ 3 (MS group).

### 2.5. Trial Interventions

Due to the pragmatic nature of the study and the need for open-label design to assess the effects of different TR protocols, we utilized available TR systems. Specifically, the HOMING Professional (https://www.tecnobody.com/) and KHYMEIA HOMEKIT VRRS ENGINE—Khymeia (https://khymeia.com/it/), both certified as Class I medical devices, facilitate the delivery of digital content for TR using task-oriented and impairment-oriented approaches for neuromotor rehabilitation in both synchronous and asynchronous modes.

To mitigate potential difficulties patients may face in using technologies due to unfamiliarity with the digital systems involved, all participants will receive comprehensive training at the clinical center, with caregivers present when possible. This training will cover device operation, internet connectivity issues, and space optimization for home rehabilitation.

Moreover, we harmonized the rehabilitation activities to establish the intervention protocol: a team of experts compared the exercises of the two TR systems available and included those exercises with the same rehabilitative characteristics in the rehabilitation protocol. Table 1 reported an example of a TR session. The harmonized protocol of TR intervention for people with PD and MS will be focused on addressing impairments and functional limitations that affect activities and participation in everyday life. In more detail, the neuromotor rehabilitation program for the TRsA group includes exercises designed to enhance functional mobility and physical capacity, following a task-oriented approach. The TRsA protocol focuses on functional activities to improve motor skills and coordination, grounded in motor learning principles to promote neuroplasticity [49]. By actively practicing real-world tasks, the brain adapts, strengthening existing neural pathways and creating new connections, which ultimately supports functional recovery. Specifically, the patient will be asked to perform task-oriented activities focused on controlling the center of mass while seated and/or during upright standing, in both static and dynamic conditions. All activities will be focused on the control of large body mass segments, head, eyes, and trunk coordination, and sensory reweighting. The difficulty level of the exercises will be adjusted in real time based on the patient’s performance, as recorded by the digital device used (Table 1).

Instead, the neuromotor rehabilitation program of the TRcA group integrates a combination of task-oriented exercises (designed to enhance functional mobility and physical ability) with impairment-oriented exercises (focused on muscle strength recovery, resistance training, and endurance improvement). The combined approach not only aims to foster neuroplasticity but also optimizes the recovery of motor functions and overall mobility. By targeting both functional tasks and muscle impairments, this program works synergistically to improve the patient’s overall physical capabilities. The exercises were specifically aimed at improving strength in muscles involved in maintaining balance and walking such as hip abductors, quadriceps, plantar flexors, and dorsal flexors. The difficulty level of the exercises will be modulated in real time based on the patient’s performance, as recorded by the digital device used (Table 1).

The characteristics of the TR intervention (TRsA and TRcA) in terms of frequency, intensity, time, and type are as follows: [a] in terms of Frequency, the TR group will undergo 5 weeks of intervention, with 4 sessions per week, delivered through a mixed model (3 asynchronous sessions and 1 synchronous session per week). Participants can choose the timing of each home TR session based on their preferences and needs; [b] regarding the Intensity, each session will be tailored to the patient’s functional abilities, ensuring progressive difficulty through the system’s feedback; [c] as regards Time, sessions will last approximately 50 min per day; [d] in terms of Type, TR activities will differ according to the neuromotor rehabilitation program proposed (TRsA vs. TRcA).

Finally, indications for home-based management following a conventional approach will be provided for the CeT group (placebo comparator). Specifically, we will implement the S.A.M.B.A. educational protocol in use at the IRCCS Fondazione Don Gnocchi in Milan (https://caditer.dongnocchi.it/samba-main/, accessed on 22 January 2025). The S.A.M.B.A. protocol consists of educational lessons (4 lessons per week for a total of 5 weeks) delivered via tablet-based content, covering five key topics: socialization, environmental factors, movement, psychological well-being, and alimentation. This structured program is designed to provide knowledge rather than exert a specific therapeutic effect.

### 2.6. Outcome Measures

Participants will undergo evaluations at three time points: baseline (T0), post-intervention (T1), and follow-up (T2) (refer to Figure 1, ‘Study Period’).

#### 2.6.1. Primary Outcome Measure

The primary outcome will be the assessment of change (T0-T1-T2) in static and dynamic balance, measured using the Mini-Balance Evaluation System Test (Mini-BESTest) [50]. The Mini-BESTest is designed to assess the underlying systems of postural control that contribute to poor functional balance. It includes 27 tasks (36 items in total) that evaluate biomechanical constraints, stability limits/verticality, anticipatory responses, postural responses, sensory orientation, and gait stability. Each item is scored on an ordinal scale from 0 to 3, with 3 indicating the best performance and 0 the worst. The total score is presented as a percentage, with higher scores reflecting better performance.

Items 7 and 8 (upright standing with eyes open on a rigid surface and with eyes close on foam, respectively) and Item 11 (walking with right/left head turns) of the Mini-BESTest will be administered in their sensorized version using three inertial measurement units (IMU) (Xsens MTw, Movella, NV, USA) attached on the lower back (L5 level) and above both lateral malleoli. Accelerometric and gyroscopic signals from the IMUs will be used to compute a set of digital metrics descriptive of gait spatio-temporal parameters (e.g., cadence, stance phase duration), stride regularity, gait symmetry, and dynamic stability for Item 11, and a set of balance indexes (i.e., sway amplitude and complexity) for the posturographic tasks involved in Items 7 and 8 [51]. See Figure 4 for a representative output of a sensorized version of the Mini-BESTest.

#### 2.6.2. Secondary Outcome Measures

Secondary outcomes will include the changes in cognitive and motor measures assessed at each evaluation time point (T0, T1, and T2).

Changes in dynamic and static balance will be measured by the Fullerton Advanced Balance Scale (FAB) [52,53], which evaluates dynamic and static balance under different situations. The FAB scale consists of 10 items that assess different aspects of balance, such as standing with eyes closed, reaching forward, turning, stepping, and standing on one leg. Each activity is scored on a 5-point scale ranging from 0 to 4, with higher scores indicating better performance. The total score can range from 0 to 40, and a score of 25 or lower suggests balance disturbances.

The modified Dynamic Gait Index (mDGI) [54] will also be used to assess changes in dynamic balance. The mDGI evaluates the ability to adapt gait to complex tasks using 8 tasks and 3 performance facets: gait pattern score (0–3), level of assistance (0–2), and time level score (0–3). The total score for each task (ranging from 0 to 8) is calculated by summing the scores from the 3 performance facets. Higher scores reflect better performance.

Changes in the perceived stability during activities of daily living will be measured by the Activities Balance Confidence scale (ABC) [55,56], while changes in the perceived fatigue in daily life will be measured by the Fatigue Severity Scale (FSS) [57,58].

The ABC is a 16-item questionnaire designed to assess an individual’s confidence in performing activities without the fear of falling or feeling unsteady. Each item is scored on a scale from 0 to 100, with higher scores reflecting greater perceived stability.

The FSS questionnaire includes nine items that assess the severity of fatigue symptoms in daily life activities. Each item is scored on a scale from 1 to 7, with higher scores reflecting greater perceived fatigue.

Changes in functional lower limb strength will be assessed using the Five Times Sit-to-Stand Test (5xSTS) [59,60]. The 5xSTS assesses functional lower limb strength, transitional movements, balance, and fall risk. The scoring is based on the time it takes for a patient to transition from a seated to a standing position and back to sitting five times. A shorter time to complete the test indicates a better outcome.

The Timed 25 Foot Walk (T25FW) [61] will be used to assess changes in mobility and leg function performance. The patient is instructed to walk 25 feet as quickly as possible, starting from one end of a marked 25-foot course. The timing begins when the instruction is given and stops when the patient reaches the 25-foot mark. The task is then repeated, with the patient walking back the same distance. The final score is the average time of the two completed trials.

Change in aerobic capacity and endurance will be measured with the 6-Minute Walk Test (6MWT). The 6MWT evaluates aerobic capacity and endurance. The distance covered in 6 min serves as the outcome measure to compare changes in performance capacity. An increase in the distance walked indicates an improvement in basic mobility. The 6MWT could be administered in its sensorized version using three IMUs on the lower trunk and ankles, as detailed by Carpinella and colleagues [62] (optional evaluation).

The general cognitive functioning will be evaluated with the Montreal Cognitive Assessment test (MoCA) [48]. The MoCA test is a highly sensitive screening battery for the detection of cognitive decline. The total score ranges from 0 to 30, and higher scores are indicative of better global cognitive performance.

The Trail Making Test (TMT) [63] and the Symbol Digit Modalities Test (SDMT) [64] will be administered to assess visuo-perceptual and attentional abilities.

The TMT is a neuropsychological test that involves visual scanning (TMT-A) and dual-task (TMT-B). It is scored by how long it takes to complete each part of the test. Longer execution times suggest lower performance.

The SDMT is a commonly used test to assess psychomotor speed. This paper-and-pencil assessment involves a substitution task using a coding key with nine different abstract symbols, each paired with a numeral. Below the key, a series of these symbols is presented, and the participant is asked to write down the corresponding number for each symbol. The score is based on the number of correct substitutions made within 90 s, with higher scores indicating better performance.

Change in the perceived level of disability will be measured by the World Health Organization Disability Assessment Schedule 2.0 (WHODAS 2.0) [65]. The WHODAS 2.0 evaluates functioning and disability levels across six domains: cognition, mobility, self-care, getting along, life activities, and participation in community activities, in line with the International Classification of Functioning, Disability, and Health (ICF). The summary scores for the WHODAS 2.0 are calculated in three steps: first, by summing the item scores within each domain; second, by summing the scores across all six domains; and third, by converting the total score into a metric ranging from 0 to 100, where 0 indicates no disability and 100 indicates full disability.

Finally, change in depressive symptoms will be evaluated with the Beck Depression Inventory (BDI-II) [66]. The BDI-II is a 21-item questionnaire, with each item rated on a 4-point scale (0–3). The score is obtained by summing the ratings for each item, with a range of 0 to 63. Higher scores indicate a greater deviation from a normal mood tone.

#### 2.6.3. Other Outcome Measures

The perceived usability of a technological system or device will be evaluated with the System Usability Scale (SUS) [67], a 10-item questionnaire. Each item is scored on a Likert scale from 0 to 5, with the total score ranging from 0 to 100. Higher scores indicate better perceived usability.

The Technology Acceptance Model-3 (TAM3) questionnaires [68] will be adopted to evaluate the user acceptance of technology across 16 domains. Scoring is based on a Likert scale (from 1 = Strongly Disagree to 7 = Strongly Agree) or a 10-point Guttman scale. Higher scores are indicative of higher acceptance of technology.

Finally, the Barthel Index [69] will be used to measure a person’s ability to perform basic activities of daily living (ADL) independently. It evaluates 10 functions, such as feeding, bathing, dressing, toileting, mobility, and stair climbing. Each activity is scored based on the level of assistance required, with a total score ranging from 0 (completely dependent) to 100 (completely independent).

### 2.7. Data Collection

The study includes the collection of clinical characteristics (e.g., the baseline motor function levels and the “ON” state during the multidimensional assessment through the Movement Disorder Society—Unified Parkinson’s Disease Rating Scale—MDS-UPDRS, Part III, [45]; the EDSS level, [47]; the age of onset, and the disease duration) and demographic information (e.g., age, education) at baseline (T0). Additionally, participants’ familiarity with and competence in using technological devices will be evaluated through an ad hoc questionnaire on technological expertise.

Data will be gathered at pre-intervention (T0), post-intervention (T1), and follow-up (T2) assessments from primary and secondary outcome measures (see the “Outcome” section of this document), with evaluations conducted by an assessor blinded to the assigned treatment. Post-intervention (T1), data on the usability, acceptability, and safety of the technological solutions will also be collected (see the “Other outcome measures” section). Safety concerns, including adverse events related to the intervention during the study, will be assessed using a clinical questionnaire.

For the TRsA and TRcA groups, data on adherence to the rehabilitation program and performance levels during exercises will be automatically recorded through the TR platform of the digital devices. In contrast, the CeT group will track adherence using a booklet.

All data collected from the three evaluation time points, along with treatment data from the digital platforms, will be entered into the REDCap database.

### 2.8. Statistical Analysis

Descriptive statistics will be employed to characterize the sample, with mean and standard deviation (SD) reported for normally distributed variables, and median and interquartile range (IQR) for non-normally distributed variables.

Data analysis will follow the modified intention-to-treat (mITT) principle [70], ensuring that all participants with available baseline data are included in the analysis. Missing data will be addressed using multiple imputation techniques to minimize bias and enhance the robustness of findings. However, we will also conduct a complete records analysis as a comparison. To test the effect of the treatment on the outcome measures, generalized linear mixed models (GLMMs) [71] will be employed. The model will evaluate changes over time (T0, T1, T2) and differences between groups (TRsA, TRcA, CeT), with outcome scores serving as dependent variables. Independent variables (fixed factors) will include time, group, and the interaction term (time × group) to capture both main effects and interaction effects. 

## 3. Discussion

The increasing demand for rehabilitation among individuals with CNDs underscores the critical need for innovative digital solutions to enhance accessibility and optimize clinical, functional, and quality-of-life outcomes. People with PD and MS are particularly reliant on rehabilitative care due to their progressive neurodegenerative nature and share common rehabilitative goals, including improving functional mobility and physical capacity. Evidence demonstrates that early, intensive rehabilitation enhances exercise tolerance, walking ability, and balance while mitigating fall risks—a critical concern for these populations [7,8,13]. However, barriers such as limited access to timely and intensive interventions persist. Digital technologies, being easily transportable, facilitate their implementation across diverse settings, including patients’ homes or other non-hospital environments. Moreover, TR is recognized as a powerful tool to strengthen primary healthcare rehabilitation services, extending access to care and overcoming logistical barriers [72].

This pragmatic trial aims to assess the effectiveness of early rehabilitative interventions delivered through a standardized TR protocol for individuals with PD and MS. Effectiveness will be evaluated not only via performance-based tests (e.g., MINI-BESTest, mDGI, …) but also by integrating patient-reported outcome measures (PROMs) such as WHODAS 2.0 and FSS questionnaire. PROMs provide invaluable insights into patients’ recovery processes, contributing to a comprehensive understanding of therapeutic success [73]. Additionally, data collected from this multicenter cohort, supported by a substantial sample size, will provide critical insights into preceding considerations, multidisciplinary assessment, and transferability evaluation. These findings will be pivotal for conducting future cost-effectiveness and acceptability conducted in accordance with the Model for Assessment of Telemedicine (MAST) framework, which is grounded in the EUnetHTA model [74].

We anticipate that TR protocols will be both usable and acceptable for individuals with MS and PD, and this rehabilitative method is expected to result in significant improvements in static and dynamic balance, functional mobility, and walking capacity in these CNDs, consistent with prior findings [22,23,25,27,29,30,33,34,35,36]. The digital tools integrated into these TR protocols are specifically selected to support intensive rehabilitation by combining task-oriented and impairment-oriented exercises.

We further hypothesize that TRcA will demonstrate superior benefits compared to TRsA, particularly for secondary outcomes such as physical capacity and functional mobility [40,41]. Based on the existing literature, task-oriented training is known to leverage neuroplasticity to enhance motor functions, including balance and mobility, while impairment-oriented approaches focus on restoring specific skills, such as strength [75].

Another noteworthy feature of TR is its ability to facilitate controlled and personalized exercises in home settings. By incorporating sensory feedback, both visual and auditory, TR actively engages patients, promoting a more interactive and effective rehabilitation experience. Moreover, TR’s flexibility allows patients to schedule rehabilitation sessions according to their daily routines and preferences, further enhancing engagement and adherence to treatment. Previous studies have demonstrated that TR approaches maintain high adherence rates, ensuring continuity of care and yielding both short- and long-term benefits in motor and non-motor functions. These include improved quality of life, greater participation in daily activities, and enhanced autonomy This adaptability is a key determinant of the structural and procedural aspects of digital treatment, playing a crucial role in achieving positive health and quality-of-life outcomes for patients [76].

Despite its benefits, TR poses challenges and limitations, particularly regarding patient unfamiliarity with technology. To mitigate the digital divide—a phenomenon where individuals lack access to or proficiency in digital tools—all participants will receive comprehensive training at the clinical center, with caregivers present where possible. This training will cover device operation, internet connectivity issues, and space optimization for home rehabilitation. Ongoing technical support will further ensure usability and address exercise-related concerns. Moreover, TR systems are expected to demonstrate not only usability but also safety when appropriately prescribed and utilized in rehabilitative care. By addressing these challenges and leveraging technological advancements, TR can provide equitable, effective rehabilitation services, transforming care for individuals with CNDs.

## 4. Conclusions

TR offers a transformative approach to addressing the rehabilitative needs of individuals with CNDs such as PD and MS. By leveraging digital tools and implementing patient-centric protocols, TR can overcome traditional barriers to care, providing equitable and effective rehabilitation services from the earliest stages of the disease. This pragmatic trial’s findings will contribute to solidifying evidence based on a robust patient cohort, highlighting the critical role of telehealth services in maintaining continuity of care at home. These findings will generate valuable data on health outcomes, laying the groundwork for future cost-effectiveness analyses. Additionally, the acceptability of the TR asynchronous modality could pave the way for further scalable solutions, such as digital therapeutics.

## Figures and Tables

**Figure 1 healthcare-13-00682-f001:**
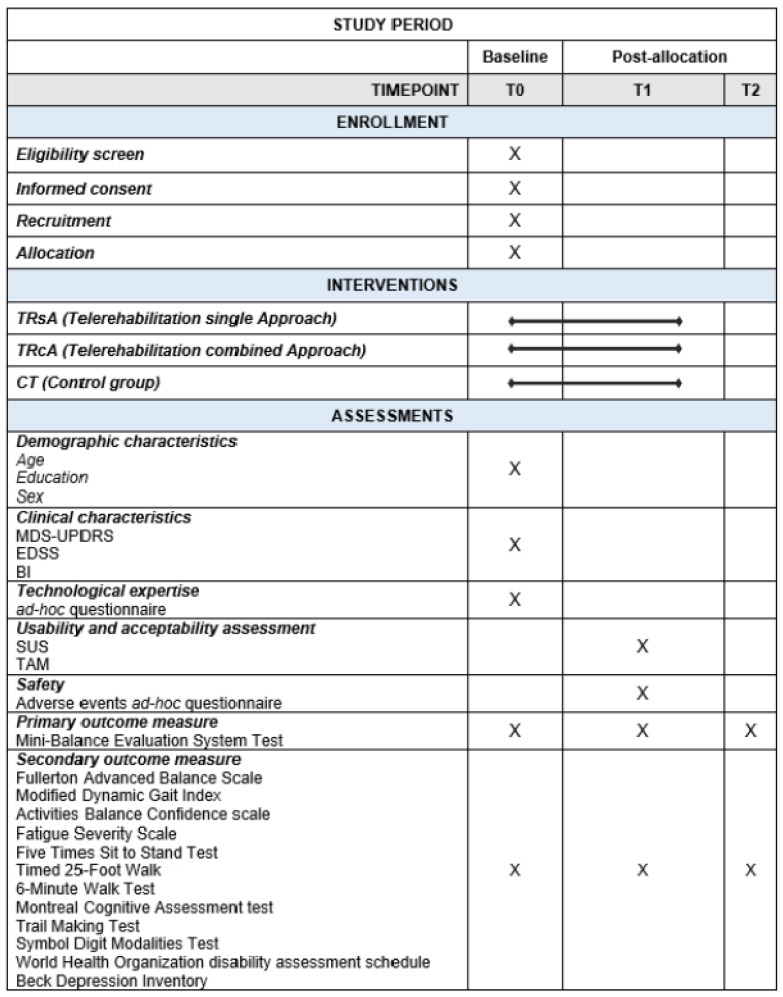
SPIRIT figure for the schedule of enrollment, interventions, and assessments in the pragmatic trial. T0 = baseline (pre-intervention phase); T1 = post-treatment assessment (5 weeks after baseline); T2 = follow-up assessment (12 weeks after the end of the treatment). MDS-UPDRS: Movement Disorder Society—Unified Parkinson’s Disease Rating Scale; EDSS: Expanded Disability Status Scale; BI: Barthel Index; SUS: System Usability Scale; TAM: Technology Acceptance Model.

**Figure 2 healthcare-13-00682-f002:**
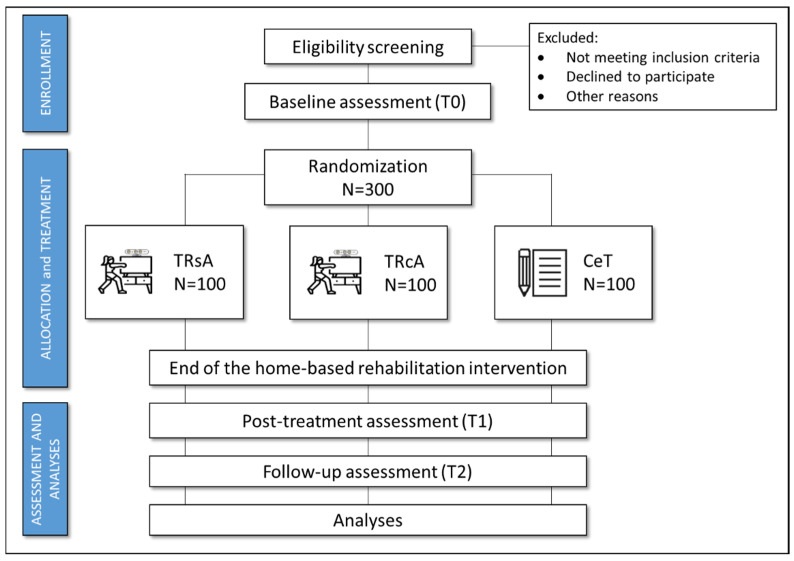
The trial work plan.

**Figure 3 healthcare-13-00682-f003:**
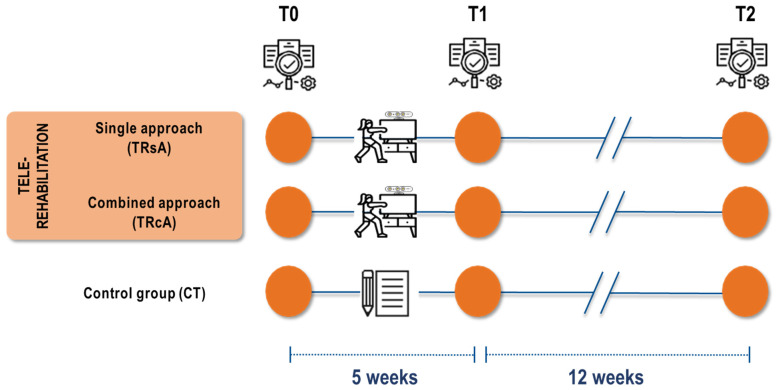
Representative figure of the clinical trials.

**Figure 4 healthcare-13-00682-f004:**
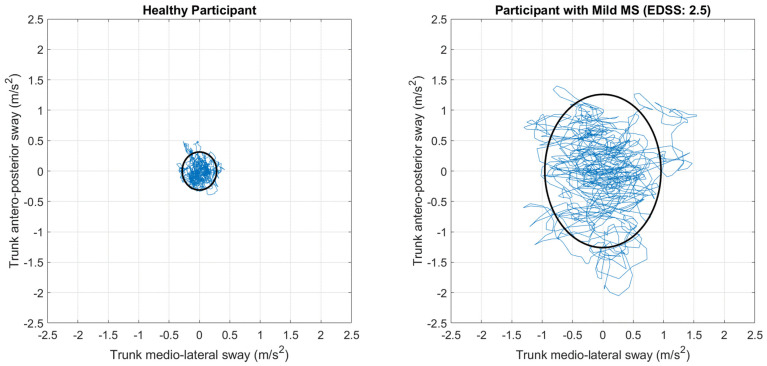
Trunk medio-lateral and antero-posterior sway, expressed as the root mean square of the acceleration detected by the IMU on lower back, in a representative healthy participant (**left panel**) and in a representative person with mild MS (**right panel**), during standing on foam with eyes close. Note the larger oscillations in the participant with MS, meaning unstable static balance.

**Table 1 healthcare-13-00682-t001:** Examples of task-oriented activities and task-oriented activities combined with impairment-oriented activities.

	Single Approach Treatment (TRsA)	Combined Approach Treatment (TRcA)
**Duration**	50 min	50 min
**Devices**	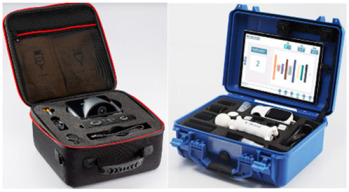	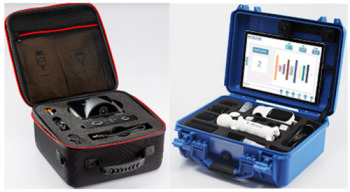
**Aim**	To improve balance and coordination 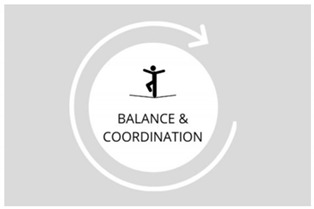	To improve both balance and coordination, and strength and resistance 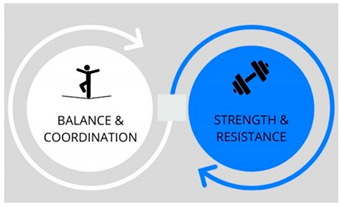
**Example of exercises**	The single approach includes exercises such as single-leg stance, standing while alternative reaching a target using the arms **E.g. EXERCISE DESCRIPTION** 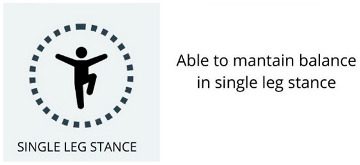	The combined approach includes exercises that simultaneously train both balance and strength such as flexion or abduction of lower limbs, and repetitive sit to stand movements from a chair **E.g. EXERCISE DESCRIPTION** 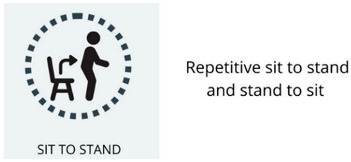

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
