# Peer review of "Timely and Personalized Interventions and Vigilant Care in Neurodegenerative Conditions: The FIT4TeleNEURO Pragmatic Trial"

_healthcare, 2025, doi:10.3390/healthcare13060682_

Round 1
Reviewer 1 Report
Comments and Suggestions for Authors
The authors present the protocol of the FIT4TeleNEURO trial, which aims to investigate the effectiveness of two different telerehabilitation protocols compared to the conventional approach in patients with Parkinson’s disease and multiple sclerosis. This is a highly interesting topic, as the potential utility of telerehabilitation -and telemedicine in general- in managing chronic neurological diseases is immense. The description of the methods is very precise and detailed. However, some minor clarifications could further improve the presentation of the work. My suggestions are as follows:
- I believe that the sentence “Among CNDs, Parkinson’s disease ..” (lines 70-71) is redundant with the previous one. I suggest eliminating one of the two sentences.
- Could you explain whether T1 refers to 5 weeks after baseline (as stated in the abstract and in paragraph 2.1) or 5 weeks after the end of the treatment (as stated in Figure 1)?
- If the Conventional educational Treatment (CeT) has a specific therapeutic effect, it should be considered a treatment comparator rather than a placebo. Could you clarify this distinction and provide any available references for the SAMBA protocol?
- Could you clarify why post-stroke patients are mentioned in line 222? Please correct it if this is a mistake.
- If the aim is to investigate the effectiveness of early rehabilitation, it is assumed that patients were diagnosed with the disease within a few years before enrolling in the study. In that case, why not also use the 2017 revised McDonald criteria for the diagnosis of MS?
- Will psychiatric disorders as an exclusion criterion be assessed through the BDI or the patient’s history?
- Rephrase the fifth exclusion criterion (at least 3 months since the last relapse) so that it clearly describes an exclusion rather than an inclusion criterion.
- Since you plan to address the issue of patients’ possible unfamiliarity with the technology, as stated in the discussion, I suggest adding this information in the methods section. I also wonder whether owning a PC and a stable Wi-Fi connection will be considered as inclusion criteria.
- Some minor linguistic adjustments are required (e.g., probably the word “trial” is missing after “pragmatic clinical” in line 138, correct the sentences “Given the pragmatic nature of the study, the inclusion criteria for all participants are be” in lines 187-188 and “The exclusion criteria for all participants are be” in line 199), and check for typos (e.g., “enrolment” in line 147).
- Make sure to explain acronyms the first time they are used, rather than repeating the explanation multiple times.
Author Response
The authors present the protocol of the FIT4TeleNEURO trial, which aims to investigate the effectiveness of two different telerehabilitation protocols compared to the conventional approach in patients with Parkinson’s disease and multiple sclerosis. This is a highly interesting topic, as the potential utility of telerehabilitation -and telemedicine in general- in managing chronic neurological diseases is immense. The description of the methods is very precise and detailed. However, some minor clarifications could further improve the presentation of the work. My suggestions are as follows:
COMMENT: I believe that the sentence “Among CNDs, Parkinson’s disease ..” (lines 70-71) is redundant with the previous one. I suggest eliminating one of the two sentences.
RESPONSE: Thanks for the suggestion, now the sentence sounds like this: “Parkinson’s Disease (PD) and Multiple Sclerosis (MS) are among the CND conditions with the highest demand for rehabilitation”.
COMMENT: Could you explain whether T1 refers to 5 weeks after baseline (as stated in the abstract and in paragraph 2.1) or 5 weeks after the end of the treatment (as stated in Figure 1)?
RESPONSE: Thank you for the observation. T1 is scheduled “5 weeks after baseline”. We have corrected the error in the caption of Figure 1.
COMMENT: If the Conventional educational Treatment (CeT) has a specific therapeutic effect, it should be considered a treatment comparator rather than a placebo. Could you clarify this distinction and provide any available references for the SAMBA protocol?
RESPONSE: The Conventional Educational Treatment (CeT) is not an active treatment but rather an informative intervention designed to provide patients with educational content about managing their clinical condition. Unlike active therapeutic interventions, CeT does not directly influence disease mechanisms or symptom progression but aims to enhance patients' understanding and self-management strategies. We have provided more details of CeT in the text regarding the educational treatment. “The S.A.M.B.A. protocol consists of educational lessons delivered via tablet-based content, covering five key topics: Socialization, Environmental factors, Movement, psychological Well-being, and Alimentation. This structured program is designed to provide knowledge rather than exert a specific therapeutic effect”..The S.A.M.B.A. protocol is currently in use at our institute and is available at the following link (Samba MAIN – SPIDERNET – FONDAZIONE DON GNOCCHI)
COMMENT: Could you clarify why post-stroke patients are mentioned in line 222? Please correct it if this is a mistake.
RESPONSE: Thank you for pointing out the error, we have removed it from the text.
COMMENT: If the aim is to investigate the effectiveness of early rehabilitation, it is assumed that patients were diagnosed with the disease within a few years before enrolling in the study. In that case, why not also use the 2017 revised McDonald criteria for the diagnosis of MS?
RESPONSE: We thank the reviewer for this observation. We have replaced the 2010 criteria with the revised 2017 criteria in the “Study Population, Recruitment, and Randomization”.
COMMENT: Will psychiatric disorders as an exclusion criterion be assessed through the BDI or the patient’s history?
RESPONSE: In our clinic, the absence of psychiatric illness is commonly assessed by reviewing the patient's psychiatric history and through clinical interviews. We specified this point in the revised manuscript.
COMMENT: Rephrase the fifth exclusion criterion (at least 3 months since the last relapse) so that it clearly describes an exclusion rather than an inclusion criterion.
RESPONSE: we rephrased the exclusion criterion as “less than 3 months since the last relapse”
COMMENT: Since you plan to address the issue of patients’ possible unfamiliarity with the technology, as stated in the discussion, I suggest adding this information in the methods section. I also wonder whether owning a PC and a stable Wi-Fi connection will be considered as inclusion criteria.
RESPONSE: as suggested, we added a paragraph in the methods section to address the issue of patients’ possible unfamiliarity with the technology: “To mitigate potential difficulties patients may face in using technologies due to unfamiliarity with the digital systems involved, all participants will receive comprehensive training at the Clinical Center, with caregivers present when possible. This training will cover device operation, internet connectivity issues, and space optimization for home rehabilitation”. Moreover, we added an inclusion criterion related to the internet connectivity issue: “Unsuitable environmental factors, such as inadequate space for rehabilitation activities or the absence of a stable internet connection”.
COMMENT: Some minor linguistic adjustments are required (e.g., probably the word “trial” is missing after “pragmatic clinical” in line 138, correct the sentences “Given the pragmatic nature of the study, the inclusion criteria for all participants are be” in lines 187-188 and “The exclusion criteria for all participants are be” in line 199), and check for typos (e.g., “enrolment” in line 147).
RESPONSE: all typos have been corrected in the revised manuscript”.
COMMENT: Make sure to explain acronyms the first time they are used, rather than repeating the explanation multiple times.
RESPONSE: We have checked and corrected all the abbreviations in the manuscript.
Reviewer 2 Report
Comments and Suggestions for Authors
Study protocol the FIT4TeleNEURO plans to address a crucial topic: rehabilitation of patients with neurological diseases using modern technologies. Due to the wide prevalence of PD and MS worldwide, often limited access to standard rehabilitation, and the high economic burden associated with these diseases, an attempt to create standardized telerehabilitation protocols is particularly valuable. The Introduction provides a concise introduction to the topic of rehabilitation in PD and MS and its challenges. The Materials and Methods provide a thorough and thoughtful description of the planned study. The study protocol is prepared by the accepted principles and does not raise any doubts. The planned tests used to assess patients are well selected. The discussion adequately presents the potential benefits and the necessity of conducting the trial.
I have a few questions and comments for the authors:
- Shouldn't you also take into account the age of onset of the disease and duration of the disease in the data you collect about patients?
- Why do you plan to use the criteria of McDonald's 2010 and not 2017?
- Will the study include only patients with relapsing-remitting MS or others as well?
- In the MDS-UPDRS scale, will patients be assessed in the "ON" or "OFF" state?
- Should the study include patients with H&Y stage 1 Parkinson's disease who do not have axial symptoms such as balance and gait disturbances?
- Please change the style of references in the text according to the journal recommendations.
- Some references do not have DOI, please complete it
Author Response
Study protocol the FIT4TeleNEURO plans to address a crucial topic: rehabilitation of patients with neurological diseases using modern technologies. Due to the wide prevalence of PD and MS worldwide, often limited access to standard rehabilitation, and the high economic burden associated with these diseases, an attempt to create standardized telerehabilitation protocols is particularly valuable. The Introduction provides a concise introduction to the topic of rehabilitation in PD and MS and its challenges. The Materials and Methods provide a thorough and thoughtful description of the planned study. The study protocol is prepared by the accepted principles and does not raise any doubts. The planned tests used to assess patients are well selected. The discussion adequately presents the potential benefits and the necessity of conducting the trial.
I have a few questions and comments for the authors:
COMMENT: Shouldn't you also take into account the age of onset of the disease and duration of the disease in the data you collect about patients?
RESPONSE: We collect data about the age of onset and the disease duration. We included this information in the “data collection” section of the revised manuscript.
COMMENT: Why do you plan to use the criteria of McDonald's 2010 and not 2017?
RESPONSE: We thank the reviewer for this observation. We have replaced the 2010 criteria with the revised 2017 criteria in the “Study Population, Recruitment, and Randomization”.
COMMENT: Will the study include only patients with relapsing-remitting MS or others as well?
RESPONSE: We appreciate the Reviewer's insightful question. Traditionally, multiple sclerosis (MS) has been classified into distinct clinical phenotypes - relapsing-remitting and progressive forms - to guide patient management, research protocols, and regulatory approvals for therapeutic interventions. However, emerging evidence (Kuhlmann et al., 2023 - The Lancet Neurology, 22(1), 78-88.) supports a paradigm shift, suggesting that MS progression should be conceptualized as a continuum rather than discrete categories. This perspective acknowledges the coexistence of multiple pathophysiological mechanisms that evolve dynamically over time and vary among individuals. In alignment with this framework, our study includes all clinical phenotypes, employing the Expanded Disability Status Scale (EDSS) as the core criterion for functional stratification and inclusion.
COMMENT: In the MDS-UPDRS scale, will patients be assessed in the "ON" or "OFF" state?
RESPONSE: all patients will be assessed in the “ON” state. We specified this in the “data collection” section of the revised manuscript.
COMMENT: Should the study include patients with H&Y stage 1 Parkinson's disease who do not have axial symptoms such as balance and gait disturbances?
RESPONSE: We thank the Reviewer for this important question. Our study includes Parkinson’s disease (PD) patients at Hoehn & Yahr (H&Y) stages 1 to 3. While balance and gait disturbances may not be clinically evident in very early PD (H&Y:1), research (Mirelman et al., 2019 - The Lancet Neurology, 18(7), 697-708) shows that subtle gait alterations, such as reduced speed, shorter step length, decreased arm swing, and increased asymmetry, occur early in the disease. Additional impairments, such as alterations in intralimb coordination, diminished range of motion at key lower limb joints during the late stance phase of gait, and increased gait variability, have also been documented (Johnson et al., 2024 - Sensors, 24(17), 5637). These changes become more pronounced under dual-task conditions due to reduced movement automaticity. The FIT4TeleNEURO protocol is designed to detect, address and treat early balance and gait impairments. Sensorized assessments like the Mini-BESTest and 6MWT enhance the sensitivity of motor impairment detection, capturing subtle deficits that may not be apparent in clinical evaluation.
COMMENT: Please change the style of references in the text according to the journal recommendations. Some references do not have DOI, please complete it
RESPONSE: The style of references has been changed according to the journal recommendations using the software Endnote.